# Comprehensive Genome-Wide Natural Variation and Expression Analysis of Tubby-like Proteins Gene Family in *Brachypodium distachyon*

**DOI:** 10.3390/plants13070987

**Published:** 2024-03-29

**Authors:** Sendi Mejia, Jose Lorenzo B. Santos, Christos Noutsos

**Affiliations:** 1Biological Sciences Department, Suny Old Westbury, Old Westbury, NY 11568, USA; 2Botany and Plant Pathology Department, Purdue University, West Lafayette, IN 47907, USA

**Keywords:** natural variation, TLP, *Brachypodium*

## Abstract

The Tubby-like proteins (TLPs) gene family is a group of transcription factors found in both animals and plants. In this study, we identified twelve *B. distachyon* TLPs, divided into six groups based on conserved domains and evolutionary relationships. We predicted cis-regulatory elements involved in light, hormone, and biotic and abiotic stresses. The expression patterns in response to light and hormones revealed that *BdTLP3*, *4*, *7*, and *14* are involved in light responses, and *BdTLP1* is involved in ABA responses. Furthermore, *BdTLP2*, *7*, *9*, and *13* are expressed throughout vegetative and reproductive stages, whereas *BdTLP1*, *3*, *5*, and *14* are expressed at germinating grains and early vegetative development, and *BdTLP4*, *6*, *8*, and *10* are expressed at the early reproduction stage. The natural variation in the eleven most diverged *B. distachyon* lines revealed high conservation levels of BdTLP1-6 to high variation in BdTLP7-14 proteins. Based on diversifying selection, we identified amino acids in BdTLP1, 3, 8, and 13, potentially substantially affecting protein functions. This analysis provided valuable information for further functional studies to understand the regulation, pathways involved, and mechanism of BdTLPs.

## 1. Introduction

Tubby-like proteins (TLPs) present in eukaryotic organisms ranging from mammals to plants are members of a highly conserved multigene family with diverse functions [1,2]. Mutations in *TLP* genes resulted in obesity, hearing loss, and retinal degeneration phenotypes in mice, while it was shown that *TLP* genes are expressed in adipocyte cells [3,4,5]. In addition, TLPs are involved in trafficking to the primary cilium, as demonstrated in mice [6,7]. The structure of a mammal TLP consists of a conserved Tubby domain at the C-terminus, whereas its N terminus is highly diverse. A significant difference to most plant TLPs is the presence of an F-box domain at their N terminus, suggesting that plant TLPs may play fundamentally independent functional roles compared to their respective mammalian counterparts.

In plants, the conserved F-box domain mediates the binding between specific substrates and components of the SCF E3 ubiquitin ligase complexes, interacting with SKP-like proteins [8]. In wheat, interactions between multiple TaTLPs and TaSKPs were shown experimentally [9]. In *Arabidopsis thaliana* (*A. thaliana*), TLP1, 3, 6, 7, 9, 10, and 11 were shown to physically interact with ASK1, 2, and 13 [10]. On the other hand, little is known about the highly conserved Tubby domain in plants. The best conserved feature of TLPs is their subcellular localization in the plasma membrane due to the binding of phosphorylated-phosphatidyl-inositol 4,5 biphosphate [PtdIns(4,5)P2] mediated via two highly conserved residues in its Tubby domain [10,11,12,13]. Upon the hydrolysis of PtdIns(4,5)P2, TLPs dissociate from the plasma membrane and translocate to the nucleus [12]. Interestingly, the same two conserved residues were shown to play a role in abiotic stress tolerance in apples [13]. Dissociation from the plasma membrane was shown also to be triggered by abiotic stresses, such as mannitol, NaCl, and H_2_O_2_ treatments in *Arabidopsis* [10].

The number of *TLP* genes in plant genomes varied from 9 to 35. The apple genome contains 9 *TLP* genes [14] compared to 11 in *Arabidopsis* [15], tomato [16], and poplar [17]; 33 in cotton [18]; 13 in cassava [19]; 14 in rice [20]; 15 in maize [21]; 22 in soybean [22,23]; 28 in rapeseed [24]; and 35 in wheat [23]. In plants, the role of TLP genes is largely unknown. In *A. thaliana*, most genes, except *AtTLP4*, are expressed ubiquitously across different organs and developmental stages [10,15]. The expression and phenotypic analysis of loss-of-function *Attlp2*, *Attlp6*, and *Attlp7* mutants showed that these genes play a role in male gametophyte development [10,25,26]. Furthermore, several *TLP* genes appear to participate in germination in response to different biotic and abiotic stresses mediated via ABA signaling [1,2,10]. Specifically, AtTLP3 and its orthologs in cucumber (*CsTLP8*) and cotton (*GhTULP34*) and *AtTLP9* negatively regulate seed germination in these plant species [10,15,27,28].

*B. distachyon* is a non-domesticated monocot species [29] that, due to its inbred nature, has retained its massive natural variation [30]. It is evolutionarily related to monocot crops, such as rice, maize, wheat, oat, sorghum, and switchgrass, which are staple for human consumption, feedstock, and biofuel production. Genomic and genetic resources are available, like the fully sequenced and assembled *B. distachyon* inbred line B21 genome, together with large genomic datasets of 54 inbred lines, including a pangenome and 16 inbred mapping populations, which offer great potential for fascinating discoveries of untapped variation [31]. Elegant high-throughput studies using two different Recombinant Inbred Line (RIL) populations (Bd1-1 × Bd21 and ABR6 × Bd21) resulted in the mapping of strong effect vernalization and flowering time loci [32,33]. 

Several studies focused on TLPs in many monocot plant species, but none included the model *B. distachyon* [9,17,20,21,23]. Here, we studied the *B. distachyon TLP* (*BdTLP*) gene family to obtain an insight into how *BdTLPs* evolved to other monocots and infer their function from high-throughput experiments [34,35,36]. In addition, we explored *BdTLPs*’ natural variation of 11 inbred lines to deduce diversity in rice and sorghum and to identify the amino acids affecting the protein structure. Through an expression profiling analysis, we showed that *BdTLPs* are expressed across different developmental stages and display differential gene expressions from darkness to light. Cis-acting elements of light, hormones, and various abiotic stresses were detected. Highly conserved to highly variable protein sequences of BdTLPs suggested that *BdTLPs* genes may have evolved different functions. This work provides the foundation for advancing our understanding of the conservation and role of *TLP* genes and informs future translational research studies on monocot species.

## 2. Material and Methods

### 2.1. Sequence Identification of TLP Genes in Arabidopsis, Rice, Sorghum, and B. distachyon

We used the publicly available Uniprot platform [37] (https://www.uniprot.org (accessed on 1 November 2023)) to find the annotated TLP genes and proteins of *Arabidopsis thaliana*, *Sorghum bicolor*, and *Oryza sativa*. The rice nomenclature was retained as noted by Bano et al. [18] and Zeng et al. [23], and all sorghum and *Brachypodium distachyon* TLPs were designated according to the rice orthologs.

We used the online PROSITE my domain image creator software (https://prosite.expasy.org/mydomains/ (accessed on 1 November 2023)) with shape option 1 and color option 4 for the proteins with an F-box, and we used shape option 6 and color option 3 for the Tubby domain [38]. Furthermore, we downloaded the *B. distachyon* Bd21 genome from the Phytozome platform [39] (https://phytozome-next.jgi.doe.gov/info/Bdistachyon_v3_2 (accessed on 1 November 2023)) to search for the *TLP* genes. For this, we used the alignment of the proteins in the three species, where TLP genes are known and mentioned above, namely *Arabidopsis thaliana*, *Sorghum bicolor*, and *Oryza sativa.* Then, two functions of the HMMER software version 3.2.2 were utilized [40,41] locally to find the TLPs in the *B. distachyon*. First, the alignment and the hmmbuild function with default parameters were used to generate the sequence profile from the aligned sequences, and then the hmmsearch function was applied with the profile against the *B. distachyon* genome 3.2 version with default parameters. Next, we used each identified protein in the Bd21 genome to perform BLAST using default parameters through the Phytozome platform (https://phytozome-next.jgi.doe.gov/blast-search (accessed on 1 November 2023)) to find the TLP genes in eleven *Brachypodium* accessions from the *Brachypodium* pangenome project. 

### 2.2. Phylogenetic and Sequence Analysis of TLP Genes

To study the phylogenies of the TLPs of the four species, we included 49 protein sequences in our analysis; AtTLP4 was excluded as it is considered a pseudogene [10,15]. We used the retrieved sequences in clustal omega alignment software (https://www.ebi.ac.uk/jdispatcher/msa/clustalo (accessed on 1 November 2023)) from EMBL-EBI [42] with default parameters to construct multiple sequence alignments (MSAs) and to retrieve the pairwise similarity matrix between pairs of protein and DNA sequences. The calculations do not take into consideration gaps in the protein sequence. MSA was visualized and further refined at four levels of conservation using the GeneDoc software version 2.7 [43] with default parameters. The aligned sequences were used in the MEGAX [44] software version 10.1.8 to calculate amino acid composition and frequencies and to build the phylogenetic tree using the Neighbor-Joining method. We tested the phylogeny using the Poisson model and 1000 bootstrap replications. We visualized the evolutionary tree via the Interactive Tree of Life (iTOL) v 6.8.2 [45] using default parameters (https://itol.embl.de/upload.cgi (accessed on 1 November 2023)). 

The DNASP software version. 6.12.03 [46] was utilized with default parameters to estimate synonymous and nonsynonymous substitution rates of the nucleotide sequences of all 49 *TLPs* and to calculate the Ka/Ks ratio of paralogous and homologous genes. 

To investigate the cis-regulatory elements, we extracted 1000 bp upstream of the transcription start site of the *B. distachyon* Bd21 genomic DNA sequence using Biomart in Phytozome [47] (https://phytozome-next.jgi.doe.gov/biomart/martview/e82b858b9cb4d6648c20f4e576f27ba2 (accessed on 1 November 2023)). Sequences obtained were scanned for cis-acting elements using the default parameters in the plantCARE platform [48] (https://bioinformatics.psb.ugent.be/webtools/plantcare/html/ (accessed on 1 November 2023)). 

### 2.3. Expression Analysis

We performed a comparative gene expression analysis at different developmental stages of the identified *TUBBY* genes in *Brachypodium*, comparing them with rice and sorghum by extracting data from the expression atlas platform [49,50]. We chose the transcripts per million (TPM) values for the comparative analysis since these allow for comparisons to be made between samples [51]. We downloaded the RNA-seq data of E-MTAB-4400 on https://www.ebi.ac.uk/gxa/experiments/E-MTAB-4400/Results (accessed on 1 November 2023), E-MTAB-4401 https://www.ebi.ac.uk/gxa/experiments/E-MTAB-4401/Results (accessed on 1 November 2023), and E-MTAB-2039 https://www.ebi.ac.uk/gxa/experiments/E-MTAB-2039/Results (accessed on 1 November 2023) for *Sorghum bicolor* BTx623, *B. distachyon* Bd21, and *Oryza sativa japonica* (Nipponbare), respectively [34]. We also used RNA-seq data from E-MTAB-7607 https://www.ebi.ac.uk/gxa/experiments/E-MTAB-7607/Results (accessed on 1 November 2023) to assay the *B. distachyon* Bd21 expression of *TLP* genes at grain development and germination [36]. Expression profiles were generated using the heatmap function from the ggplot2 package [52] with default parameters. The Z-score was calculated as (gene expression value in a gene of interest)—(mean expression across all samples)/standard deviation and was used to interpret gene expression data [53]. It was plotted on a scale from −2 to +2, with negative values related to downregulation and positive values related to upregulation, respectively. We also explored available RNA-seq data E-MTAB-6445 to score differential expressions of *BdTLPs* during photomorphogenesis https://www.ebi.ac.uk/gxa/experiments/E-MTAB-6445/Results (accessed on 1 November 2023) and hormone-induced gene expression data from DDBJ with accession number PRJDB2997 [35]. 

### 2.4. Natural Variation and 3D Modeling of BdTLPs’ Variant Proteins

We extracted coding and protein sequences of eleven *B. distachyon* accessions used as parents for constructing RIL populations based on their percent identity from the *B. distachyon* pangenome website [54] https://phytozome-next.jgi.doe.gov/blast-search (accessed on 11 December 2023). Variations in coding and protein sequences were studied using a pairwise comparison of Ka and Ks, similarly to the ortholog calculations. We used the Ka/Ks ratio as a marker to identify amino acids of interest in the MSA of accessions. All amino acid sites were counted from the Bd21 protein sequence position. The identified amino acid variants in the BdTLP1, 3, 8, and 13 proteins were used to build their 3D structure models. Each amino acid change is denoted by the Bd21 amino acid position, followed by the variant amino acid, e.g., in BdTLP1, the Bd21 conserved C amino acid at position 61, mutated to the S amino acid is denoted as C61S. The amino acid sequences were submitted to the RobeTTAFold website to generate 3D protein images [55] with default parameters. The US align online software (https://zhanggroup.org/US-align/ (accessed on 1 November 2023)) (default parameters) overlaps the tertiary structures of the BdTLP1, 3, 8, and 13 variants [56]. The effect of mutated residues on protein stability was predicted using DDmut [57] (https://biosig.lab.uq.edu.au/ddmut/ (accessed on 1 November 2023)) and DUET [58] (https://biosig.lab.uq.edu.au/duet/ (accessed on 1 November 2023)) for both software default parameters applied. 

## 3. Results

### 3.1. Classification and Phylogenetic Analysis of Arabidopsis thaliana, Oryza sativa, Sorghum bicolor, and Brachypodium distachyon TLP Genes 

Using HMMER software version 3.2.2, we identified 12 full-length putative *TLP* genes in *B. distachyon* spread across all five chromosomes, specifically one in chromosome 5, two in chromosomes 1 and 4, three in chromosome 3, and four in chromosome 2 (Appendix A). We also retrieved previously identified TLPs from the Uniprot platform https://www.uniprot.org (accessed on 1 November 2023), specifically 10 from *A. thaliana*, 13 from *S. bicolor*, and 14 from *O. sativa*. All BdTLPs are paralogous to OsTLPs. An ortholog of OsTLP11 is found in sorghum only, while OsTLP12 is rice-specific (Appendix A). The majority of TLPs in all three species consisted of an F-box in the N terminus, whereas four proteins, namely BdTLP13, OsTLP13, SbTLP13, and AtTLP8, only consisted of the Tubby domain (Appendix A). 

To gain an insight into the evolutionary relationships of *B. distachyon* TLPs, we constructed a phylogenetic tree using the full-length amino acid sequences of *Brachypodium*, rice, sorghum, and Arabidopsis. The 49 TLPs formed six groups (Figure 1). 

Specifically, group 1 consisted of TLP4, TLP6, TLP1, and TLP5 of rice, sorghum, and *Brachypodium*, respectively, clustered with AtTLP1, 5, and 10 (Figure 1). Group 2 consisted of TLP2 and TLP3 of rice, sorghum, and *B. distachyon* with no *Arabidopsis* proteins present. Group 3 included AtTLP2 and 6, and TLP 8 of the three poaceae species, whereas group 4 consisted of AtTLP3, 9, and 11 and TLP7 and 14 of rice, sorghum, and *B. distachyon*. Group 5 consisted of TLP9 and TLP10 of rice, sorghum, and *Brachypodium* that cluster together with the rice and sorghum TLP11, the rice-specific TLP12, and more distantly to AtTLP7. Finally, in group 6, the proteins only consisted of the Tubby domain, such as AtTLP8, BdTLP13, SbTLP13, and OsTLP13, are clustered together. *B. distachyon* TLPs grouped either with the respective *O. sativa* TLP orthologs (TLP3, 4, 8, 9, and 14) or with the respective *S. bicolor* orthologs (TLP2, 7, and 10).

To study the similarity of TLPs, we calculated the similarity matrices of all TLPs against all other homologous and orthologous TLPs using the clustal omega software (https://www.ebi.ac.uk/jdispatcher/msa/clustalo (accessed on 1 November 2023)) as described in the Materials and Methods Section. The within-similarity *B. distachyon* proteins, except BdTLP13, was, on average, 58.39%, whereas BdTLP13’s similarity to all other *B. distachyon* proteins was 26.63%. Of the different group pairs, BdTLP1-BdTLP5 shared 81.32% similarity, whereas BdTLP9-BdTLP10 shared 65.73% similarity (Appendix A). Interestingly, certain BdTLPs shared higher similarity to rice proteins, while others shared a higher similarity to sorghum proteins (Appendix A). Specifically, BdTLP5 and BdTLP13 shared higher similarities to the respective sorghum orthologs, namely 86.61% and 79.18%, respectively. All other BdTLPs shared higher similarity with the respective rice orthologs, ranging from 76.14% for OsTLP6 to 92.36% for OsTLP1. These data suggest that TLPs in monocot species are highly conserved compared to the more distantly related dicot species, such as *Arabidopsis*.

To quantify the molecular evolution of *TLP* genes, we estimated Ka and Ks and computed the Ka/Ks ratio to infer the selection pressures (Appendix A). Most orthologous and homologous *TLP* gene pairs are under purifying selection (Ka/Ks < 1). Of the 26 gene pairs that exhibited diversifying selection (Ka/Ks > 1), six were between homologous gene pairs, such as *AtTLP8* to *AtTLP5*, *11*, and *3*; *OsTLP13* to *OsTLP11* and *12*; and *BdTLP13* to *BdTLP3.* Of the other 20 gene pairs, 12 were between *AtTLP8* and other rice, sorghum, and *Brachypodium TLP* genes; however, the strongest diversifying selection of *AtTLP8* was observed for *AtTLP3* and *AtTLP11*. Strong diversifying selection was also found for *OsTLP13* to *BdTLP13* and *SbTLP9*, and less strong diversifying selection was found for *SbTLP13*, *OsTLP12*, *OsTLP11*, and *AtTLP8*. *BdTLP13* was another gene with a Ka/Ks ratio higher than 1, especially with *BdTLP3* and *OsTLP13*, as mentioned. The pair of genes under the strongest diversifying selection were *BdTLP10* and *OsTLP10*. These data suggest that the orthologous and paralogous *TLP* genes may have developed diverse function(s) since they are under strong selection. 

### 3.2. Multiple Sequence Alignment Analysis and Amino Acid Composition Bias of AtTLPs, OsTLPs, SbTLPs, and BdTLPs

Multiple sequence alignments (MSAs) are used to locate conserved regions and to provide structural, functional, and phylogenetic information of proteins and/or domains. The MSA of the 49 TLPs revealed that group 5 is the most diverse than the other groups, consisting of a two-amino-acid insertion (serine and glycine) in the F-box domain that is also present in AtTLP9 and AtTLP11 and highly variable 14 residues at the end of the F-box that are dominated by hydrophobic residues. Moreover, group 5 proteins are missing about 90 amino acids in the middle part of the Tubby domain, resulting in a reduced protein length (Appendix A). These 90 highly variable residues are part of two insertions, split by a highly conserved stretch of amino acids. In the large insertion, residues are present in groups 1–4, whereas in the small insertion, residues are present in groups 1 and 2 only. The residues in the smaller insertion are primarily proline, serine, and glutamine. The impacts of these F-box and Tubby domain insertions and the role of specific amino acids remain to be elucidated. 

We then focused on residues of known impact. In previous studies, it was shown that the conserved F-box and Tubby domains localize to the nucleus and plasma membrane, respectively [10,11,12,13]. Two amino acids, K187 and R189, were revealed to have crucial roles in the AtTLP3 plasma membrane tethering via specific phosphoinositide binding [10,11]. In groups 1–5, except for OsTLP12, which shows K187N, these residues are conserved or substituted with similar amino acids such as K187R (AtTLP9) and R189K (AtTLP5, OsTLP8, BdTLP8, and SbTLP8), suggesting similar protein subcellular localization (Appendix A). In group 6, these amino acids are replaced by serine, threonine, or glycine residues, suggesting that they may have abolished plasma membrane tethering. 

The amino acid sequence determines the protein structure and its folding properties. To test the conservation of the TLP sequence, we explored their amino acid composition (Figure 2 and Appendix A). Amino acid composition deviations from the average were observed for at least one amino acid in all 49 proteins. SbTLP13 displayed the highest number of amino acids deviating from the average (12 amino acids), followed by OsTLP13 (11 amino acids), AtTLP2 (10 amino acids), AtTLP6, and AtTLP8 (9 amino acids) and BdTLP13 (8 amino acids). We searched for amino acid compositional bias patterns at three levels: (1) a high and low residue numbers conserved among the same orthologs, (2) a high or low residue number at the species level, and (3) major differences between orthologs. At level 1, BdTLP13, OsTLP13, and SbTLP13 displayed the highest numbers of alanine residues and the lowest numbers of phenylalanine and serine residues. BdTLP4, SbTLP4, and OsTLP4 displayed the highest numbers of cysteine residues, while BdTLP6, SbTLP6, and OsTLP6 displayed the lowest numbers of arginine residues. BdTLP3, SbTLP3, and OsTLP3 displayed the highest numbers of aspartic-acid residues and serine residues. The highest numbers of serine residues were also detected in SbTLP2, BdTLP2, and OsTLP2, suggesting that group 2 proteins may have been selected for some shared but also diverse functions. OsTLP8, BdTLP8, and SbTLP8 displayed the highest numbers of leucine residues and the lowest numbers of tyrosine residues. OsTLP9, SbTLP9, and BdTLP9 displayed the highest numbers of tyrosine and glutamic acid and the lowest numbers of isoleucine residues. Group 5 members, except AtTLP7 and SbTLP11, exhibited the highest numbers of tryptophan residues. The conservation among orthologs may indicate selection for specific gene functions.

At level 2, *Arabidopsis* proteins were among the most biased. AtTLP2, 7, 8, and 11 exhibited high numbers of threonine residues; AtTLP5, 6, 7, and 8 displayed high numbers of lysine residues; AtTLP6, 8, 10, and 11 displayed high numbers of leucine residues; and AtTLP3, 6, 7, and 8 displayed high numbers of asparagine residues. AtTLP2, 5, 6, and 8 displayed low numbers of arginine residues. *S. bicolor* proteins SbTLP3, 9, 10, 11, and 14 exhibited high numbers of glutamine residues, while SbTLP1, 5, 7, and 10 exhibited low numbers of leucine residues. Rice proteins OsTLP9, 10, 11, and 13 exhibited high numbers of glutamic acid residues. Although the amino acid compositional bias in various *B. distachyon* proteins is apparent, itis not as widespread as that in the other species. 

At level 3, we looked at contrasting patterns occurring within the same group, such as some members with high numbers of residues and the rest with low numbers of the same residue. AtTLP8 displayed the lowest number, and BdTLP13, SbTLP13, and OsTLP13 displayed the highest numbers of alanine residues, whereas BdTLP13, SbTLP13, and OsTLP13 displayed the lowest numbers of serine residues compared to AtTLP8. Given that alanine is hydrophobic and serine is polar and serves as a phosphorylation site, AtTLP8 may exert different structures and/or functions than its orthologs.

### 3.3. Cis Elements in BdTLP Promoters and Gene Expression during Developmental Stages

The transcription regulation of genes is modulated by cis elements; therefore, searching for cis elements may infer gene function. A total of 731 cis elements were identified by scanning 1000 bp upstream of the translation start site of the 12 *BdTLPs*, ranging from 44 in *BdTLP9* to 89 in *BdTLP4* (Figure 3, Table 1, and Appendix A). A total of 1/3 of the 731 cis elements predicted were CAAT-box and TATA-box. Of the other 2/3 of the identified cis elements, 9% was involved in light, 8% in hormones, 5% in environmental stresses, 2% in development, and 42% in processes other than the above. 

Of the 68 unique cis elements, the majority are responsive in abiotic stresses such as anaerobic-inducing element (ARE), low-temperature response element (LTR), a drought and salt response element (MYC), and an MYB binding site involved in drought (MBS) anoxic induction and wound and defense responses (WUN-motif and TC-rich repeats). The hormone-related cis elements found were the ABA response element (ABRE) identified in all but *BdTLP2*, *6*, and *9*; gibberellin-responsive elements (GARE-motif and P-box); methyl jasmonate elements (CGTCA-motif and TGACG-motif); and the salicylic acid responsive element (TCA-element). Notably, we also identified many light responsive elements such as G-box, which was detected in all but *BdTLP2*, *9*, and *14*, and GATA-motif, which was detected in *BdTLP2*, *4*, *5*, *7*, *8*, and *14*. Other light-responsive elements predicted were Gap-box, GT1-motif, I-box, MRE, Sp1, the TCCC-motif, and the TCT-motif. The CAAT-box, MYB, TATA-box, and Unnamed 4 were present in all gene promoters. Also, cis elements involved in seed-specific regulation (RY-element) were identified in *BdTLP2* and *6*, and the meristem expression CAT-box was distributed in the promoter of *BdTLP2*, *4*, *6*, *9*, and *14*, respectively. These data indicate that *BdTLPs* are involved in diverse pathways in response to environmental, hormonal, and light stimuli.

Based on the cis-acting elements analysis, *BdTLPs* may be involved in light, hormone, and development responses. To investigate the expression patterns of the *B. distachyon TLPs* across development, we analyzed three publicly available RNA-Seq datasets [34,36]. The three poaceae species’ (*B. distachyon* B21, *O. sativa japonica* var. Nipponbare, and *S. bicolor* BTx623) *TLP* expressions were analyzed across comparable vegetative (leaf), reproductive (early inflorescence (EI), emerging inflorescence (EMI), anther, and pistil), and seed (seeds at 5 days and 10 days after pollination (DAP), embryo, and endosperm) stages (Figure 4b–d) [34]. Remarkable conserved expression patterns were observed in all orthologs. The majority of genes were upregulated during the reproduction stages (EI, EMI, anther, and pistil) and downregulated in the vegetative tissues and late stages of seed development in an expression gradient from reproductive to seed stages. Interestingly, most *BdTLPs* genes were upregulated in anthers (9 out of 12 genes), with *BdTLP3*, *BdTLP4*, and *BdTLP10* being highly upregulated and *BdTLP4* and *10* being downregulated in all other stages. *OsTLPs* and *SbTLPs* were highly upregulated in pistils. *OsTLP10* displayed a unique expression profile, highly upregulated at EI, anther, and embryo, while *SbTLP10* and *SbTLP8* shared similar profiles, being upregulated at EI, pistil, and seed 5DAP. In seeds, 5DAP, and embryo, some genes are still highly upregulated, while other genes are already downregulated.

To further explore the putative expression of *BdTLPs*, we retrieved and analyzed the public expression RNA-seq data, as described in Materials and Methods, from young grains to germinating grains and young seedlings (Figure 4a) [36]. Most *BdTLP* genes are upregulated in germinating grains and young seedlings. Of these genes, *BdTLP1*, *3*, *5*, and *14* are downregulated in later developmental stages, whereas *BdTLP2*, *7*, *9*, and *13* remain highly expressed during early reproductive stages. *BdTLP4*, *6*, *8*, and *10* are only upregulated at reproductive stages. All genes are downregulated in mature grains. 

To identify which *BdTLPs* are involved in photomorphogenesis, we analyzed the differential expressions of *BdTLP* genes after a transition from constant darkness to constant light (https://www.ebi.ac.uk/gxa/experiments/E-MTAB-6445/ (accessed on 1 November 2023) in Bd21 seedlings. Four *BdTLPs* responded to the transition from darkness to light. Of these four genes, *BdTLP4* was downregulated, and *BdTLP3*, *BdTLP7*, and *BdTLP14* were upregulated (Appendix A). The data during photomorphogenesis are in frame with the identified cis elements (Figure 3 and Appendix A). In another transcriptome analysis of various hormones in Bd21 [35], *BdTLP1* was found to be strongly responsive to 10 µM ABA after 1 h and less responsive in 30 µM methyl jasmonate (meJA) after 1 h. Moreover, *BdTLP5* was detected in the low stringency datasets of the auxin, salicylic acid (SA), and meJA responsive genes, whereas *BdTLP14* was identified in the low stringency dataset of the SA-responsive genes. The expression data confirmed that *BdTLPs* have diverse functional roles.

### 3.4. Natural Variation DNA and Protein Sequences and 3D Structures of BdTLPs

We explored the natural variation in 11 *B. distachyon* accessions that are parents of RIL populations. For each BdTLP, we constructed MSA (Appendix A) to study their conservation. BdTLP5 is the most conserved gene with a single amino acid polymorphism. We determined three categories of proteins: proteins with few amino acid changes, such as BdTLP1, 4, 5, and 6 (category 1); proteins with few amino acid changes and small or large insertions and deletions, such as BdTLP2, 3, and 8 (category 2); and proteins with high residue variation and insertions/deletions in one or more lines, such as BdTLP7, 9, 10, 13, and 14 (category 3). The least conserved proteins across parental lines either had few amino acid changes and large deletions over 100 amino acids (BdTLP7, 9 and 13) or many amino acid changes and large deletions (BdTLP10 and 14). For example, Tek-4 and Bd3-1 exhibited deletions of 169 and 126 amino acids and variations in 13 amino acids in BdTLP7, while in BdTLP10, the same lines exhibited deletions of 188 and 79 residues and a variation in 101 amino acids, of which 87 were in Bd3-1. These three categories largely follow the evolutionary relations found in Figure 1, where the most conserved group 1 BdTLPs belong in category 1, groups 2 and 3 belong in category 2, and the most diverse groups 4, 5, and 6 belong in category 3. 

To study sequence conservation among lines, we extracted similarity matrices of the different coding and protein sequences (Appendix A) using clustal omega as described in the Materials and Methods Section. The averaged similarity in the BdTLP1-9 proteins ranged from 98.90% in BdTLP8 to 99.96% in BdTLP5, whereas at the coding sequence, it ranged from 99.25% in *BdTLP7* to 99.85% in *BdTLP5*. In BdTLP10, 13, and 14, the similarity ranged from 93.29% in BdTLP10 to 97.50% in BdTLP13 proteins, and in the coding sequence, it ranged from 93.93% in *BdTLP10* to 98.12% in *BdTLP13*. Collectively, these results may indicate that BdTLP5 is an indispensable gene in *B. distachyon*.

Natural genetic variation may impact gene function and enhance plant fitness [59]. Most substitutions were synonymous, and there were fewer changes to non-synonymous amino acid substitutions. The ratio of synonymous/non-synonymous changes is an indicator of diversifying, purifying or no selection. We assayed Ka/Ks within the DNASP software (version. 6.12.03), as described in the Materials and Methods Section, in all possible pairs of parental lines on all 12 *BdTLPs* coding sequences (Appendix A). Although the vast majority exerted purifying selection or no selection, a Ka/Ks higher than 1 was observed in multiple pairs for *BdTLP1*, *BdTLP3*, *BdTLP8*, *BdTLP13*, and only one pair was observed in *BdTLP6* and *BdTLP9* (Appendix A). In *BdTLP1*, positive selection was observed between Koz-3 and Bd3-1, Bd21-3, Bd21, Bd2-3, Bd30-1, and RON2 accessions, where a cysteine residue at amino acid 61 was changed to a serine residue (Appendix A). The C61S resides are seen in the beginning of F-box. In *BdTLP3*, positive selection was observed between Koz-3, Bd21-3, and Bd3-1 and Bd30-1, ABR6, Bd1-1, and RON2 and between Bd2-3 and Bd21 to Bd30-1 accessions (Appendix A). From the alignments, Koz-3, Bd21-3, Bd2-3, Bd3-1, and Bd21 have conserved tyrosine, valine, and leucine residues in 323, 346, and 380 amino acids compared to conserved phenylalanine, alanine, and proline residues, respectively, which were found in the remaining accessions (Appendix A). In BdTLP8, four amino acids in Bd3-1 and one in Bd30-1 at residues 96–99 were found to fall into the F-box. In BdTLP13, the Tubby only consisting of a domain protein, positive selection was detected in accessions Bd30-1 to Tek-4; Bd29-1 and Bd1-1; and Tek-4 to Koz-3, RON2, Bd21, and Bd2-3. Bd30-1 differs from Tek-4, Bd29-1, and Bd1-1 at amino acid 352, where a glutamine residue is changed to arginine. Bd-1 and Tek-4 have a conserved proline residue that is converted to a threonine in Koz-3, Bd2-3, Bd21, and RON2. These data suggest that the above genes are under strong positive selection and may exhibit diverse functions at different *Brachypodium* genetic backgrounds due to single residue changes.

We searched the conservation of the above amino acids in the rest of the TLP sequences (Appendix A). The cysteine residue C61 in the BdTLP1 F-box was replaced by serine in BdTLP3 and 5; AtTLP2 and 3; OsTLP1, 5, and 6; and SbTLP3 and 6. It was replaced by glycine in BdTLP2; OsTLP2, 3, and 11; SbTLP2; arginine in BdTLP9, 10, and 14; OsTLP9 and 10; and SbTLP7, 9, and 10. It was replaced by glutamine in BdTLP7; OsTLP7 and 14; and SbTLP14. And it was replaced by asparagine in BdTLP4, 6, and 8; OsTLP4 and 8; SbTLP4, 5, and 8; and AtTLP1, 5, and 10. The tyrosine residue Y323 in BdTLP3 was replaced by isoleucine in BdTLP1, 2, 4, 5, and 13 and their orthologs, by serine in BdTLP6, and by aspartic acid in BdTLP8, while it fell into the big deletion of Tubby domain in BdTLP7, 9, 10, and 14. Phenylalanine was only found in OsTLP3. The valine residue V346 in BdTLP3 was also found in BdTLP1 and 2 and in OsTLP2 and 12, but replaced by alanine in BdTLP4, 6, 8, 9, and 10; by serine in BdTLP7, 13, and 14; and by all their orthologs, except SbTLP13 and proline in BdTLP5 and its orthologs. The leucine residue L380 was found in BdTLP3 and SbTLP2, whereas proline was found in BdTLP5 and 9 and their orthologs, namely BdTLP2, AtTLP1 and 10, SbTLP1, and OsTLP11 and 12, whereas glutamine, serine, threonine, and asparagine were found in BdTLP1, 7, 10, and 14, respectively. This residue was not present in BdTLP4, 8, and 13 and their orthologs. The valine residue V96 was conserved in all BdTLPs and their orthologs, except OsTLP11. The valine V97 was conserved in groups 1–5 and replaced by glycine in group 6. The alanine residue A98 was conserved in 26 proteins and replaced by serine in BdTLP1 and its orthologs, BdTLP9 and its orthologs, SbTLP11, and OsTLP11, and it was replaced by cysteine in BdTLP7 and its orthologs and arginine in the group 6 proteins. The cysteine residue C99 was also highly conserved, except in BdTLP5 and its orthologs, where it was replaced by phenylalanine, and in SbTLP2, it was replaced by serine. The threonine residue T132 in BdTLP13 corresponded to C61 in BdTLP1. The valine residue V185 in BdTLP13 was conserved in all group 6 proteins and replaced by alanine in BdTLP2, 7, 8, 9, 10, and 14; by serine in BdTLP1, 3, 5, and 6; and by lysine in BdTLP4, with alanine and serine being conserved among the orthologs. The glutamine residue Q352 in BdTLP13 was replaced by lysine in BdTLP1, 2, 4, 5, 6, 8, and 14 and their orthologs; by threonine in BdTLP3, 9, and 10 and their orthologs, except SbTLP3 and 11; and by aspartic acid in BdTLP7 and its orthologs. Notably, most amino acid changes are conserved among orthologs, and some are expected to be tolerated, while others will have a profound impact on the protein structure and function. 

The typical structure of TLPs consisted of a β-barrel with 12 anti-parallel strands with a central hydrophobic α-helix. To identify the impact of the amino acid changes on the protein structure, we generated 3D models of BdTLP1, 3, 8 and 13. The overlap structures of Bd21 and the modified protein are shown in Figure 5. All protein structures predicted by RoseTTAFold were altered at the N terminal, and the most conserved were BdTLP3 V346A and L380P. Furthermore, to gain insights on the effects of the missense mutations in protein stability, we ran the DUET and DDmut algorithms. All mutated residues were predicted to destabilize the proteins. The strongest effects were observed in Y323F in BdTLP3; V96G and C99G in BdTLP8; and V185G in BdTLP13 (Appendix A). These results show that using natural variants is an effective strategy in the identification of candidate residues with a strong effect in protein structure, stabilization, and, ultimately, function. 

## 4. Discussion

In plants, TLPs consist of an F-box and a Tubby domain. Several F-box proteins act as positive or negative regulators in plant stresses, involved in hormonal and light signaling, and in developmental processes from seed germination to floral development [60,61,62,63,64]). In *B. distachyon*, 12 putative TLPs were identified (Appendix A) that were classified into six groups, five of which consisted of an F-box/Tubby, and one group that only consisted of Tubby domain, respectively (Figure 1). Group 2 did not consist of *Arabidopsis* TLPs, suggesting that the functional evolution of monocot TLP2 and TLP3 genes may have taken place. The results of our phylogenetic analyses are in accordance with the findings by Zeng and others [19,22,23,24]. We furthermore used the Ka/Ks ratio as an indicator of selective pressure [17] and found strong diversifying selection between the paralogs and ortholog TLPs (Appendix A). Other studies have not found diversifying selection between the paralogs or orthologs of TLPs [23,24], except for BnaC05g20780D to Bo5g043430.1 in *Brassica napus* [24].

Hormone, light, abiotic, and biotic environmental stress-responsive cis elements were predicted in *BdTLPs* (Figure 3 and Appendix A). Our findings on cis elements are consistent with the published data in other plant species [14,16,18,19]. In the hormone-induced transcriptome profiles, *BdTLP1* was one of the four hundred and forty-five differentially expressed ABA-responsive genes after 1 h of 10 µM of ABA [35]. *BdTLP1* and *5* were also detected at a low-stringency meJA dataset after 1 h of 30 µM of meJA [35], suggesting that other *BdTLPs* may be expressed after 1 h or at different concentrations. Although many studies showed that TLP genes are stress-induced, especially in the ABA signaling pathway and in cold, salt, and drought stresses [10,11,15], the role of *TLP* in light has not been revealed. Four genes displayed differential gene expressions during photomorphogenesis (Appendix A), implying that the predicted cis elements are not only confirmed by other studies, but furthermore can be used as sources to reveal additional *TLP* gene functions. 

Gene expression across comparable vegetative, reproductive, and seed parts was screened in rice, sorghum, and *B. distachyon* (Figure 4b–d) [34]. Our findings for the sorghum and rice *TLP* gene expression profiles were more similar and fit well with the trend proposed by Davidson et al. [34]. In a second dataset [36], *BdTLPs* were expressed across developmental stages, from as early as germinating grains and young seedlings in the cases of *BdTLP1*, *2*, *3*, *5*, *7*, *9*, *13*, and *14* to only at reproductive stages in the cases of *BdTLP4*, *6*, *8*, and *10* (Figure 4a). These data, together with the photomorphogenesis gene expression profiles, suggest that some *TLP* members have broad functions, while others may have more specialized functions. 

Two residues that are highly conserved in mammals as well in plants play central roles in plasma membrane binding [10,12,20]. Studies in AtTLP3 and Tubby have demonstrated that mutating these sites to alanine abolishes the PM binding [10,11,12]. We found that, except OsTLP12 K→N, these two positively charged residues are substituted by similar amino acids, e.g., K→R in AtTLP9 and R→K in AtTLP5, OsTLP8, BdTLP8, and SbTLP8, suggesting similar protein subcellular localization (Appendix A). In group 6 proteins, these residues are replaced by threonine (K→T) and serine or glycine (R→S or R→G) residues, respectively, suggesting that group 6 proteins have probably abolished plasma membrane binding [11,12]. Three other conserved residues within the mammalian Tubby phosphoinositide binding sites N310, N348, and R363 [12] are conserved in plant species, studied here as glutamic acid, proline, and phenylalanine residues, respectively, with the last being the least conserved. A plausible mechanism of plasma membrane binding is the release and transport to the nucleus, so incorporating more key amino acids in plants is of outmost importance. 

Natural variation is the key to informing strategies in the development of abiotic stress-tolerant varieties. Through the screening of natural variation present in *B. distachyon* accessions, we deduced amino acids that were either conserved or diverse at the orthologs or paralogs (Appendix A). Amino acid substitutions can have a positive or negative impact on the protein structure and stability. We interrogated eleven amino acid substitutions in four BdTLPs and found altered protein structures and stability in varying degrees (Figure 5, Appendix A). In apple, similar findings suggested that differences in the 3D protein structures may lead to the functional diversification of MdTLPs [14]. As such, we anticipate that BdTLP3 Y323F, BdTLP8 V96G and C99G, and BdTLP13 V185G and Q352R will have a strong effect on the protein structure and possibly function. In BdTLP13, Q352 coincided with the Y464 in the Tubby protein and N366 in AtTLP3 [11], a residue that is important for the nuclear translocation of the Tubby protein that is mediated via the insulin phosphorylation of Y464 [65]. Site-directed mutants will inform the role of the identified variant amino acids in BdTLPs under salinity and drought stress conditions.

## Figures and Tables

**Figure 1 plants-13-00987-f001:**
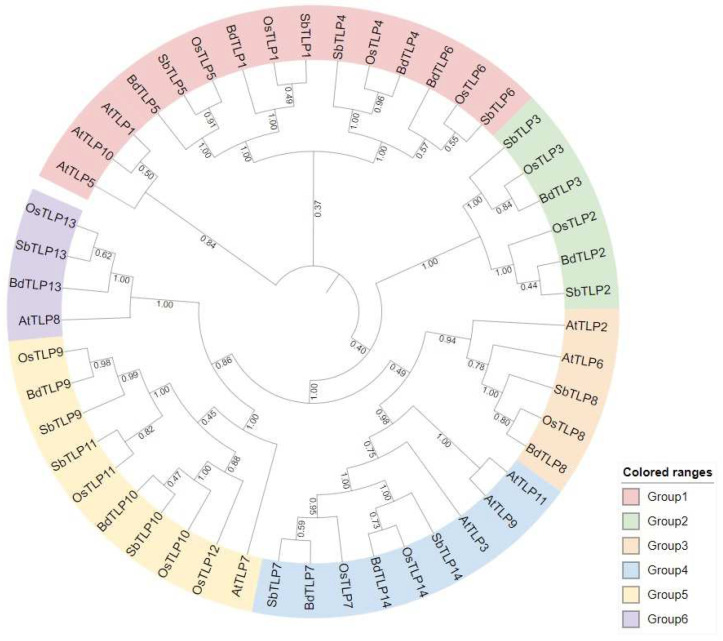
Evolutionary relationships of 10 *A. thaliana*, 12 *B. distachyon*, 13 *S. bicolor*, and 14 *O. sativa* TLPs. Five groups were identified.

**Figure 2 plants-13-00987-f002:**
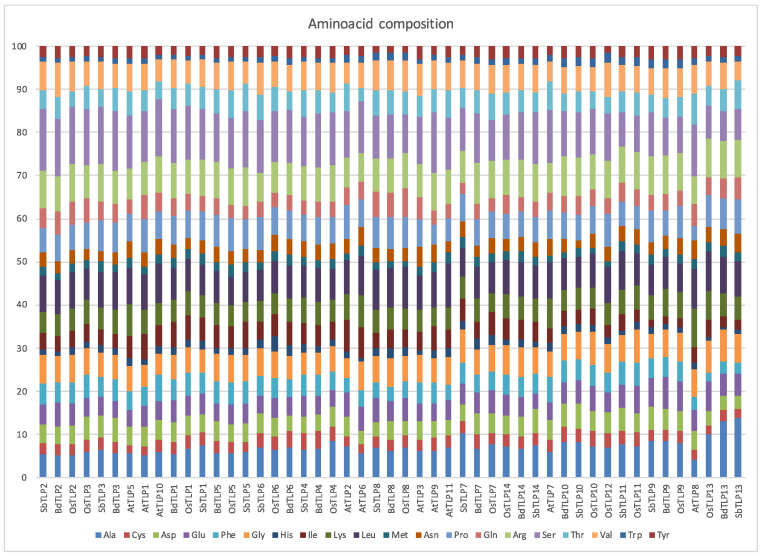
The amino acid compositions across all 49 TLPs. Each amino acid is displayed with a different color.

**Figure 3 plants-13-00987-f003:**
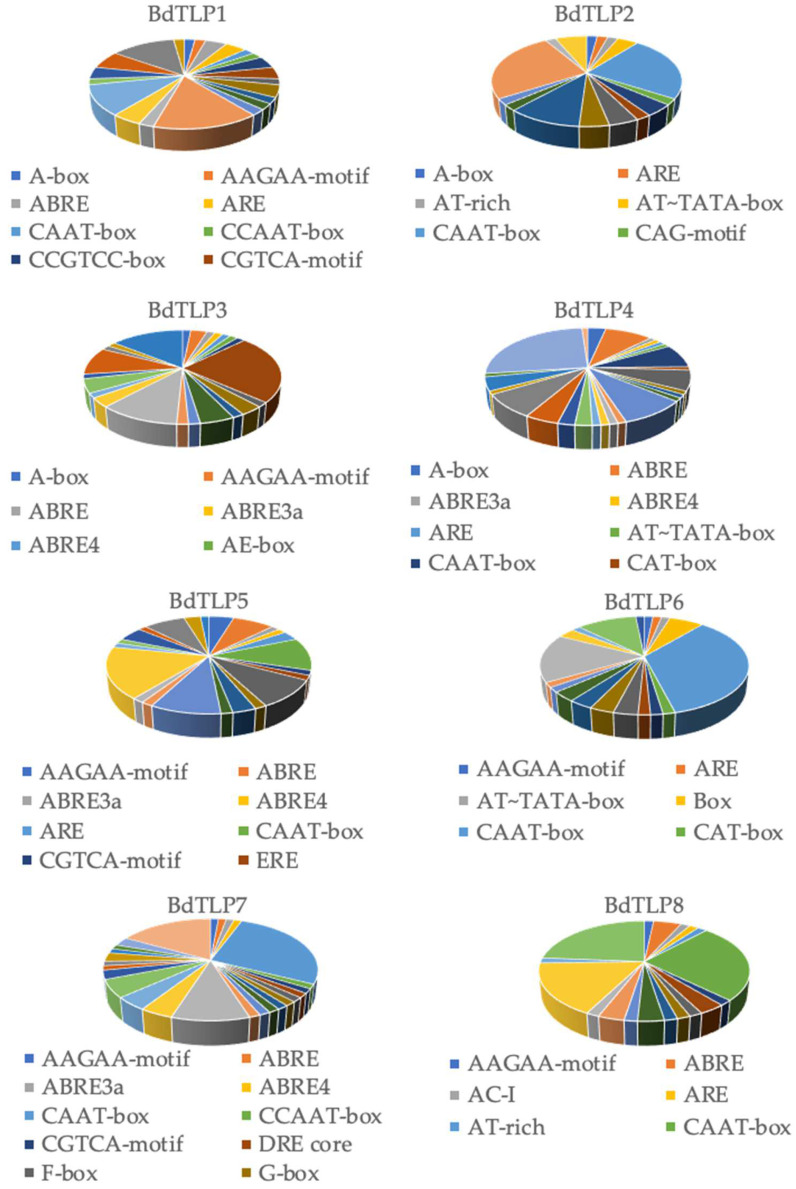
Cis elements involved in growth, development, light, stress, and hormonal responses.

**Figure 4 plants-13-00987-f004:**
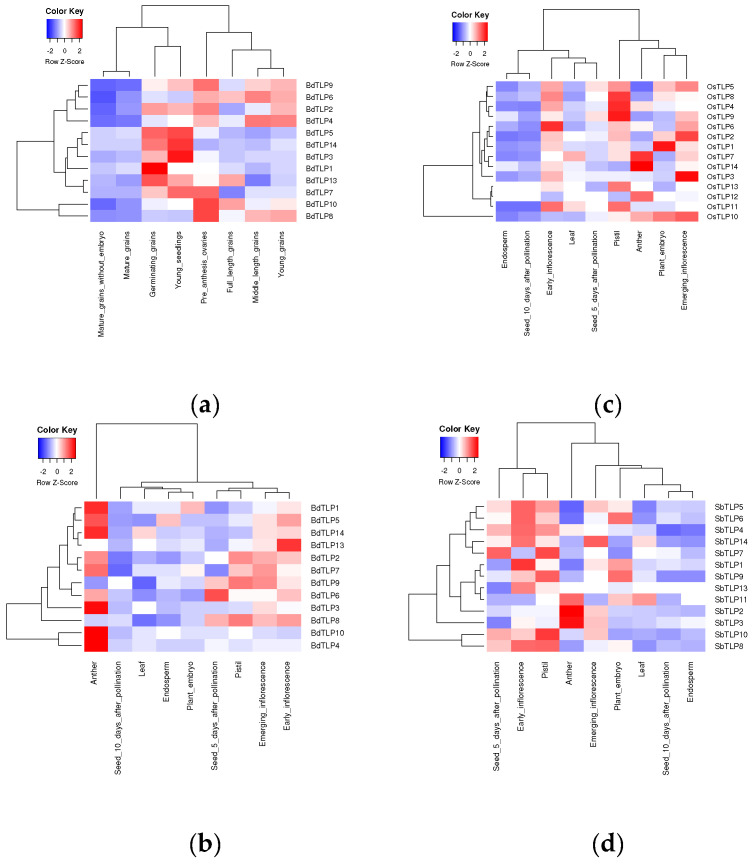
Expression analysis of (**a**,**b**) BdTLPs and of (**c**) OsTLPs and (**d**) SbTLPs at different developmental stages. Red z-score indicates upregulation and blue downregulation of gene expression.

**Figure 5 plants-13-00987-f005:**
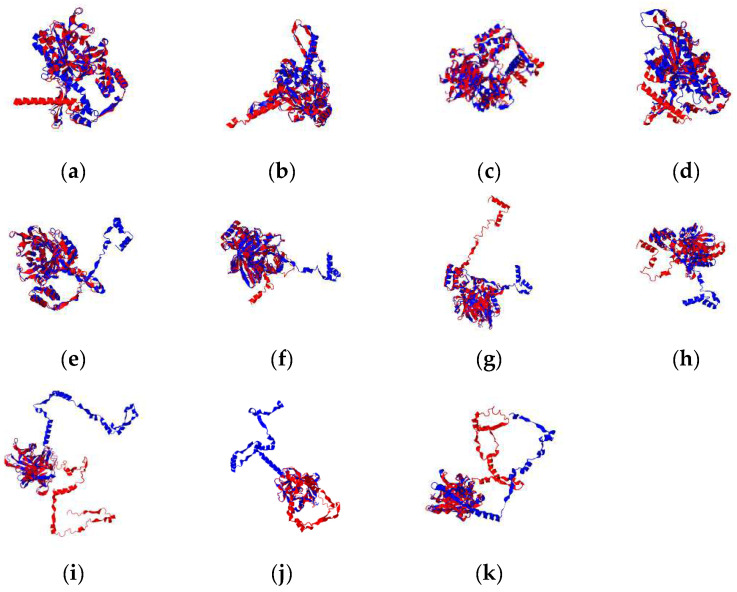
Overlap protein structures of wild-type Bd21 and mutated proteins. Blue denotes Bd21 structure and red denotes mutated 3D structure. Three-dimensional structures of (**a**) BdTLP1 C61S, (**b**) BdTLP3 Y323F, (**c**) BdTLP3 V346A, (**d**) BdTLP3 L380P, (**e**) BdTLP8 V96G, (**f**) BdTLP8 V97G, (**g**) BdTLP8 A98G, (**h**) BdTLP8 C99G, (**i**) BdTLP13 T132P, (**j**) BdTLP13 V185G, and (**k**) BdTLP13 Q352R are shown.

**Table 1 plants-13-00987-t001:** The members and types of cis-acting elements in the promoters of BdTLPs.

	Light	Hormone	Environmental Stress	Developmental	Promoter	Other	Total
**BdTLP1**	6	7	3	0	6	24	**46**
**BdTLP2**	2	1	1	3	23	17	**47**
**BdTLP3**	5	7	2	1	21	27	**63**
**BdTLP4**	13	11	3	1	14	47	**89**
**BdTLP5**	9	10	4	0	18	23	**64**
**BdTLP6**	2	4	3	4	32	18	**63**
**BdTLP7**	5	7	5	0	23	31	**71**
**BdTLP8**	5	4	3	0	25	22	**59**
**BdTLP9**	1	1	3	2	17	20	**44**
**BdTLP10**	9	4	4	1	9	41	**68**
**BdTLP13**	3	2	4	0	27	28	**64**
**BdTLP14**	3	4	3	3	29	11	**53**
**Total**	**63**	**62**	**38**	**15**	**244**	**309**	**731**

## Data Availability

Data are contained within the article and Appendix A.

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
