# Peer review of "Comprehensive Genome-Wide Natural Variation and Expression Analysis of Tubby-like Proteins Gene Family in Brachypodium distachyon"

_plants, 2024, doi:10.3390/plants13070987_

Round 1

Reviewer 1 Report

Comments and Suggestions for Authors

Comments and Suggestions for Authors

Dear Authors,

It is my pleasure to review the manuscript entitled “Comprehensive Genome-Wide Analysis of Tubby-Like Proteins (TLP) Gene Family in Brachypodium distachyon, Oryza sativa, Sorghum bicolor and Arabidopsis thaliana” a research article submitted to MDPI Journal, Plants. Authors of this manuscript identified and characterized several Tubby-Like Proteins (TLP) in Brachypodium distachyon and compared other TLP in several other plants. They have characterized physical and chemical properties, gene structure, phylogeny, and expression patterns through a series of bioinformatic analysis. Huge information for these genes has been provided. Overall, the experiments, they performed, are well and the results are convincing. Thus, the presented results take up an important topic consistent with the profile of the Journal.

However, I have some suggestions, which might improve the manuscript to make important to the wider readers.

Abstract: Good organization with results order, but not matching with title. Title contain B. distachyon, O. sativa, S. bicolor and A. thaliana. But abstract does not contain any of them. Remove this inconsistency.

L19; Two major groups (A and B). it is not giving any specific meaning

Introduction:

-Introduction should be more specific with rationale of the study. Elaborate clearly, why this research is necessary. Here huge bioinformatic experiments have been done, those should have some reflection in introduction. Not necessarily important to present animal research if not specifically corelated.

-Based on abstract and introduction, title should not contain other than B. distachyon,

Materials

-The word “we” have been used several times. Can be reduced by using passive form

-L121-124; need reference

L131; only RILs is good enough

L142; what is NGL?

Results

-L150-153; rephrase this sentence with reduced “TLP genes”.

-L151; A. thaliana, O. sativa…… is enough

-Only “Arabidopsis” may not be italic

-L159-160; best fit in discussion

-L163, and L150-152; make linear

-Point 3.2 is too wordy. Need to reduce to keep full attention for the readers

-Need high resolution of Fig. 3.

L320; what are those data sets are?

-L336-342; Is this also silicon experiment? Or real? If so, need description in the Material and Methods.

-None of the gene’s expression has been validated by lab real experiment which is in some extend lowering the strength of this article.

Author Response

I have included my answers to attached document

Reviewer 2 Report

Comments and Suggestions for Authors

As currently written, it is not clear why this study was performed. The paper is written in a very descriptive way and fails to generalize results. I have the following recommendations:

Introduction:

There are no aims described in the current version of this paper. Later the readers find that the authors "identified 12 putative BdTLP genes and analyzed their evolutionary relationships to rice, sorghum, and Arabidopsis. BdTLPs are expressed across different developmental stages and display differential gene expression from darkness to light." - that's as much as we get from the introduction, which is different from the title of the paper.

Methods:

This section is clear but the authors need to detail the conditions used in each program. 

Results:

Many results appear without any relation with the description stated in the methodology. Many results - in fact, a large majority- are stated as SI files.

Discussion:

The discussion is overall a repetition of results without a clear generalization of results.

Minor comments:

Genes and species should be italicized throughout the paper.

Author Response

I have included my answers to the below attachment

Reviewer 3 Report

Comments and Suggestions for Authors

The present research on Genome-Wide Analysis of Tubby-Like 2 Proteins (TLP) Gene Family in Brachypodium distachyon, Oryza sativa, Sorghum bicolor and Arabidopsis thaliana provides a good insight and comparison within the 4 species studied and mainly in B. distachyon.

The abstract is a reflect of the research and the results are clearly explained.

The introduction gives a good overview of theknowledge on the topic, is well referenced and presents in detail the objectives of the research.

The material and method section should be completed as long as there are several information mentioned in the results section which do not appear in this section.

The results are clearly presented and the discussion is complete with good references. Only you should avoid mention some material and method here.

Author Response

(The authors gave the same response as above.)

Round 2

Reviewer 2 Report

Comments and Suggestions for Authors

Thank you for time and efforts in answering to my previous comments/doubts, and for improving this second version. Congratulations.

Reviewer 3 Report

Comments and Suggestions for Authors

Authors have now provided a reviewed version of the document, adjusting the material and method section and result section to avoid redundancies and data missing.

the other parts were already fine for me